# Preparation of Chitosan Stacking Membranes for Adsorption of Copper Ions

**DOI:** 10.3390/polym11091463

**Published:** 2019-09-06

**Authors:** Xiaoxiao Zhang, Xuejuan Shi, Liang Ma, Xuan Pang, Lili Li

**Affiliations:** 1Key Laboratory of Automobile Materials, Ministry of Education, and College of Materials Science and Engineering, Jilin University, Changchun 130022, China; xiaoxiao17@mails.jlu.edu.cn (X.Z.); xjshi17@mails.jlu.edu.cn (X.S.); liangma17@mails.jlu.edu.cn (L.M.); 2Key Laboratory of Polymer Ecomaterials, Changchun Institute of Applied Chemistry, Chinese Academy of Sciences, Changchun 130022, China; xpang@ciac.ac.cn

**Keywords:** chitosan, sandwiched stacking structure, copper ions, adsorption

## Abstract

Chitosan (CS) stacking mats with excellent performance for adsorption of copper ions (Cu(II)) in wastewater were fabricated by alternating electrospinning/electrospraying. The hierarchical structure of the stacking membranes was designed by CS micro-hemispheres sandwiched between CS fibers. The CS stack membranes prepared by the electrospinning technology could effectively increase the specific surface area, and thus, facilitate the adsorption of copper ions. CS stacking membranes with three layers reached adsorption equilibrium within 60 min, and had a maximum absorbance of 276.2 mg/g. The absorbance performance was superior to most of the reported CS adsorbents. Compared with CS fiber mats which were dominated by CS chemical structure during adsorption, the stacking structure of CS membranes contributed to the high efficient capability, and exhibited the multilayer adsorption behavior. This study may develop a promising method for the design of environmentally-friendly natural polymer adsorbents to remove Cu(II) in wastewater.

## 1. Introduction

Copper is a heavy metal universally contaminating wastewater, which is mainly from industries such as electronics, metal processing, steel production, fertilizer, pesticides and so on [1,2,3,4,5]. Copper ions (Cu(II)) tend to accumulate in living organisms. Long-term contamination of Cu(II) will cause damage to the human health, such as vomiting, hypotension, kidney damage and even death [6,7]. The effective elimination of Cu(II) pollutants from aqueous solution has attracted concerns from environmental investigators. Currently, the main methods for removing heavy metals are chemical precipitation [8], coagulation-flocculation [9], ions exchange [10] and membrane adsorption [11]. Among them, adsorption is a simple and highly efficient way of dealing with heavy metals in aqueous solution [12,13].

At present, there are various adsorbents containing carbon materials [14], biomaterials [15], chelating materials [16], metal oxide nanoparticles [17,18] and so on. However, most adsorbents, such as carbon materials, are limited in their applications because of their high cost, easy agglomeration, difficulty in processing and low adsorption efficiency, etc. Therefore, adsorbent materials with wide sources, high removal efficiency and low cost have become a research hotspot. Chitosan (CS) is an alkaline natural biomaterial with abundant natural resources. It is non-toxic, odorless, biodegradable and environmentally friendly [19]. CS molecular chains have a great quantity of amino and hydroxyl functional groups which can generate the complexes by cooperating with heavy metal ions [20]. In current research, an effective way to obtain CS-based adsorbents is described, i.e., by improving the specific surface area and adsorption performance via the subtle design of the adsorbent morphology [21,22,23,24,25,26].

Electrospinning/electrospraying are simple and effective techniques for designing different morphologies of materials which can fully utilize the superior adsorption characteristics of materials by obtaining high specific surface area, porosity, permeability, et al. [23]. Vu et al. fabricated CS/magnetic (Fe_3_O_4_)/iron hydroxide (Fe(OH)_3_) microspheres with the adsorption amount for As(III) of 8.47 mg/g [24]. Wang et al. prepared CS/poly(ethylene oxide) (PEO)/permutit (PT) fibers mat which had greatly improved adsorption efficiency for the Cr(VI) removal [25]. However, the adsorption performance of the material is not only connected with the adsorption site, but also with the adsorption space. Tu et al. loaded CS or CS-REC spheres on the PS fibers membranes surface, and the adsorption performance remained 134 mg/g [26]. Although the hierarchical structure with the larger surface area and adsorption space improved the adsorption capability compared to single layer adsorbent, a lot of interesting research on hierarchical structure designing has focused on the addition of multiple components in recent work.

It was believed that the adsorption behavior was affected by both the hierarchical structure and the distribution of the added component. Based on the above research, CS as the only component was used to design the stacking adsorbent with multi-morphology.

In this study, to study the effect of hierarchical structures of a CS-based adsorbent on the adsorption behaviors of Cu(II), a pure CS stacking membrane with different sandwich structures was prepared by the electrospinning/electrospraying method. CS stacking membranes stabilized by sodium carbonate (Na_2_CO_3_) solution were used for Cu(II) adsorption. Most importantly, the effects of the thickness and morphology of interlayers on Cu(II) adsorption were investigated in detail. This research may provide theoretical guidance for designing natural polymer adsorbents.

## 2. Experiment

### 2.1. Materials and Chemicals

CS (M_w_ = 200 kDa, 90% deacetylated) was supplied by Aladdin Industrial Corporation, Shanghai, China. Hexafluoroisopropanol (HFIP) was provided from Shanghai Darui Finechemical Company (Shanghai, China). Formic acid (FA), acetic acid (AA), sulfuric acid (H_2_SO_4_), sodium hydroxide (NaOH), sodium carbonate (Na_2_CO_3_) and copper sulfate (CuSO_4_·5H_2_O) were obtained from Beijing Chemical Works, China. All the reagents were used as received without further purification.

### 2.2. Properties of Electrospinning and Electrospraying Solutions

The physical properties of different electrospinning and electrospray solutions are shown in Table 1. CS could only be dissolved in an acidic solvent such as FA which had a poor volatility and generated a large repulsive force in an electric field. It was difficult for CS to form continuous fibers. HFIP had favorable volatility and compatibility with FA. Therefore, HFIP was added to the FA to obtain a FA/HFIP mixed solvent system for electrospinning solutions. CS fibers (CS-F) were electrospun using HFIP/FA as solution with various volume ratio. CS microspheres (CS-M) were electrosprayed using AA as solution. The properties of all the prepared CS solutions are shown in Table 1. When the HFIP content decreased, the conductivity and viscosity of CS-F solutions increased. For the electrospraying CS-M solutions, as the CS concentration increased, the viscosity and conductivity of solution increased.

### 2.3. The Fabrication of Monolayer Membrane

CS fibrous (CS-F) membranes were fabricated using electrospinning method. In order to obtain continuous CS nanofibers, the optimum ratio of the electrospinning solution (3% *w*/*v*) was determined through dissolving CS powder in different volume ratios of HFIP/FA mixed solvent of 9/1, 6/1, 3/1 (*v*/*v*), respectively. 3% CS electrospinning solutions in a 5 mL syringe were fed with flow rate of 0.8 mL/h on a syringe pump at the applied voltage of 18 kV. CS microspheres (CS-M) was electrosprayed by changing the concentration of CS from 2 wt% to 4 wt% with 90% AA as solvent. The experimental conditions of electrospraying were as follows: feed rate of 0.4 mL/h, voltage of 18 kV and the operation distance of 15 cm.

### 2.4. The CS Stacking Memebranes by Alternated Electrospining/Electrospraying

The CS stacking membranes were fabricated via the alternated electrospinning/electrospraying technology at 25 °C and 10% relative humidity. CS fibrous membranes was collected on an aluminum foil by electrospinning for 5 h, and labeled as CS-L1.

For the preparation of CS stacking membranes, firstly, CS fibers were collected by electrospinning for 2 h. Then electrosraying CS porous concave hemispheres were collected on the surface of the CS fibrous membrane for 1 h, and the membrane with two-layers was marked as CS-L2. CS-L3 was obtained by electrospinnig CS nanofibers on CS-L2 for 2 h. According to this principle, the CS fibers and concave hemispherical particles were alternately deposited layer by layer. CS membranes with n-layer stacking structure was marked as CS-Ln. The preparation procedure of the adsorption membranes was illustrated in Scheme 1.

### 2.5. Nanofibers Neutralization Treatment

Because of the soluble protonated amino groups (NH_3_^+^) in CS molecules, the prepared CS stacking mats were immersed in 1 mol/L Na_2_CO_3_ solution to enhance the stability in aqueous solution by charge neutralization [19,27]. After treatment for 2 h, the CS mats were repeatedly washed with distilled water to reach the neutral pH environment, and then vacuum dried at 40 °C for 12 h.

### 2.6. Characterization of CS Stacking Membranes

The solution’s viscosity was determined by viscosimeter (NDJ-1, Yutong, Shanghai, China). The solution’s conductivity was determined by conductometer (DDS-11A, Shengci, Shanghai, China). The above tests were performed 3 times at 25 °C. The morphology of CS stacking membranes was tested by FE-SEM (JEOL 6700F, Tokyo, Japan) at a working voltage of 5 kV. The average diameter of the fibers and hemispheres were determined by measuring in 100 different areas using Image J analysis software. The chemical structures of the CS stacking membranes were analyzed by FT-IR spectroscopy (FTIR-4100, JASCO, Shanghai, China) from 400 to 4000 cm^−1^. The surface area, pore volume and pore size were characterized by N_2_ adsorption and desorption using a Brunauer–Emmett–Teller analyzer (BET, Autosorb-iQ2, Quantachrome Instruments, Shanghai, China). The composition on the surface of the CS membranes was examined by XPS (ESCALab220i-XL, VG Scientific, MA, USA). The membranes were measured in the range of 10–70° (2θ) with the Cu Kα radiation at 40 kV. Inductively coupled plasma atomic optical spectrophotometer (ICP-OES, Ultima Expert, Horiba, Paris, France) was employed to survey the content of the heavy metal ions in the solutions.

### 2.7. Adsorption Experiments

The adsorption solution of Cu(II) was prepared by putting 2.0 g CuSO_4_·5H_2_O into 500 mL deionized water. For adsorption experiments, 10 mg adsorbent was soaked in 15 mL Cu(II) solution with 100 rpm shaking speed at 25 °C. The pH values were controlled by 0.1 mol/L H_2_SO_4_ or 0.1 mol/L NaOH solution. The adsorption process lasted for 24 h with original Cu(II) concentration of 200 mg/L at room temperature. The adsorption isotherms of Cu(II) were constructed via initial concentrations in the range of 50–500 mg/L. The adsorption amount *q* (mg/g) of Cu(II) for each CS adsorbent was determined based on the Equation (1):(1)$$q=\frac{C_{0}-C_{e}}{m}\;\times \;V$$
where *C*_0_ and *C_e_* (mg/L) were the original and the balance concentration of Cu(II) in solution, respectively. *V* (L) was the volume of the solution and *m* (g) was the weight of the initial membrane.

## 3. Results and Discussion

### 3.1. Morphology of Electrospun/Electrosprayed CS Layers

Figure 1 shows the morphology of electrospun CS fibers with different volume ratios of HFIP/FA as solvents. As shown in Figure 1a,b, CS fibers displayed a string of beads structure with HFIP/FA volume ratio of 3/1 (*v*/*v*), the average diameter was 49 ± 13 nm. When HFIP/FA volume ratio increased, resulting in the decrease of the viscosity and conductivity (Table 1), the bead on CS fibers gradually decreased and the fiber diameter increased. The decrease of the conductivity was the primary reason for the increase of the fiber diameter. In particular, in the case of HFIP/FA with volume ratio of 9/1, the smooth fibers without beads were observed [28], and the fibers diameter increased to 127 ± 24 nm, which would be used as CS fiber layers of CS stacking membranes.

The CS interlayers with higher specific surface areas were designed for further improving Cu(II) adsorption space. For this purpose, the effect of the concentration on the morphology of electrospraying CS was studied in detail (see Figure 2). When CS content was 4 wt%, the electrospun CS displayed a string of beads configuration. The beads became larger accompanying the thinner fiber with CS content decreased. The porous concave hemispheres with average diameter of 1.22 ± 0.56 μm could be obtained at 2 wt% of CS content [29,30,31]. The porous concave hemispheres mats had high specific surface area, which were expected to provide more adsorption site and space. Therefore, CS porous concave hemispheres were selected for the following studies.

### 3.2. The Preparation of CS Stacking Membranes

The CS stacking membranes were obtained by alternating electrospinning/electrospraying means. The electrospun CS-L1 for 5 h were used as the reference adsorbent of CS staking membranes (see Figure 3a). Figure 3b showed the morphology of CS hemispheres, which was used as interlayers on the CS fibers for 2 h. The architectures for hemispheres upon the fibers could be observed apparently. Based on these experiments, CS-L3 stacking membranes were obtained from alternated fiber/hemispheres/fiber layers. The cross-section image of CS-L3 membrane was revealed in Figure 3c. The sandwich structure with CS hemispheres as interlayers could be observed obviously. In the same way, CS-L5 were developed from alternated hemispheres/fibers multilayer sandwich structure with 3 layered fibers and 2 layered hemispheres. As shown in Figure 3d, the multilayer structure was built up apparently in the way that two hemispheres layers sandwiched by three fibers mats respectively.

### 3.3. The Stability in Aqueous

For the improvement of the stability in aqueous solution, all the CS adsorption membranes were treated in a 1 M Na_2_CO_3_ solution for deprotonation. Figure 4a,b showed the SEM micrograph of the CS-L1 and CS-L2 after neutralization. The morphologies of the CS membranes had not change much. The mean diameter for fibers and hemispheres increased to 250.2 ± 59.5 nm and 1.69 ± 0.38 μm respectively. The enlargement of CS-L2 in Figure 4b manifested the interconnection structure between the hemisphere and fibers, which might derive from the partly dissolved hemispheres and fibers in unevaporated solvent.

The structure of stabilized CS mats were examined by FT-IR and XPS spectra. Compared with FT-IR spectra of CS, the amino cations peak at 1714 cm^−1^ disappeared after treated by Na_2_CO_3_ (Figure 5). This indicated that amino group was deprotonated. XPS spectrums of N1s for CS mats before and after neutralization were shown in Figure 5b. The peak of N1s at 397.9 eV for CS mats had a slight shift after neutralization, suggesting that ammonium ions were converted to amine groups [20]. The treated CS membrane could present in the aqueous medium stably after deprotonation treatment [32].

### 3.4. BET Analysis

The BET data of all the CS adsorption mats after stabilization treatment are listed in Table 2. CS stacking membranes with sandwich structures (CS-L3 and CS-L5) had a higher pore diameter, pore volume and specific surface area than the pure CS fiber membrane (CS-L1). These results demonstrated that CS stacking membranes had more adsorption sites and better permeability compared with CS monolayer fiber membrane.

### 3.5. Adsorption Perfomance

Figure 6a illustrates the influence of adsorption time on the adsorption ability of Cu(II) by CS adsorption mats. The adsorption tests were performed under 200 mg/L Cu(II) concentration and pH of 6 at room temperature. As shown, the Cu(II) adsorption performance of all membranes improved as the contact time increased, until reaching the saturation adsorption quantity of the adsorbents. At the initial adsorption period (*t* < 20 min), the adsorption amounts increased dramatically due to vast free adsorptive sites and high concentration of the metal ions. Afterwards (*t* > 20 min), the adsorption rates began to decrease, indicating that the adsorption capacity for all membranes gradually reached the equilibrium; this was ascribed to the consumption of adsorptive sites and the decrease of metal ion concentration. In addition, the adsorption capacity of CS-L3 (185.4 ± 2.5 mg/g) and CS-L5 (188.7 ± 3.6 mg/g) were both higher than CS-L1 (175.0 ± 3.5 mg/g), indicating that CS stacking membranes could supply more adsorption space for Cu(II) compared with CS monolayer membranes.

The adsorption rate could be seen from the slope of the curve in Figure 6a. In the initial phase of adsorption, the adsorption rates for all adsorbents were list in the order: CS-L3 > CS-L1 > CS-L5. The membranes of the CS-L1 and CS-L5 reached adsorption equilibrium after 180 min of adsorption, while the CS-L3 adsorption membrane with three-layered sandwich structure attained adsorption equilibrium after 60 min. It was believed that the high porosity and optimal thickness for aqueous penetration of CS-L3 resulted the shorter equilibrium time [33].

The water flux of all CS adsorption membranes at a pressure of 0.095 MPa were tested (Figure 6b). The results exhibited that the CS-L3 adsorption membrane had the highest water flux in all CS adsorbents, which further explained that CS-L3 had excellent permeability. The CS-L3 adsorption membrane had a simpler preparation process than the CS-L5 adsorption membrane and a higher adsorption efficiency than the CS-L1 adsorption membrane. Therefore, the CS-L3 adsorption membrane was selected for further research.

The pH values of metal ions solution was an affecting factor on both adsorbent and adsorbate. Cu(II) adsorption performances were investigated in pH range of 2–6 using original Cu(II) concentration of 200 mg/L (see Figure 7). The adsorption capacity of Cu(II) increased when pH value ranged from 2 to 6. The maximum adsorption capacity of CS-L1 and CS-L3 was achieved at pH value of 6. At pH values lower than 6, the low Cu(II) adsorption capacity was caused by electrostatic repulsion between Cu(II) and the protonation of the amino group of CS. When pH values were high, the decrease in the adsorption capability was attributed to the adsorption site reducing. This reduction was caused by the sedimentation of metal complexes formed on the surface of adsorption membranes [34].

The effect of initial Cu(II) concentration on adsorption performance and the removal rate was also observed. In Figure 8a, the adsorption performance of the adsorption membranes improved with the increase of Cu(II) concentration, but the adsorption rate decreased. As seen from Figure 8b, CS-L3 showed nearly 100% and 93% removal rate at the concentration of 100 mg/L and 200 mg/L, respectively. Nevertheless, the removal rate of mats reduced as the Cu(II) original concentration improved. This behavior could be ascribed to the limited adsorption sites [35]. In addition, CS-L3 mats showed higher adsorption ability and removal rate than CS fibrous membrane.

### 3.6. The Equilibrium Isotherm and Rate Constant Studies

The kinetics adsorption behaviors of the heavy metal ions were investigated by pseudo-first-order and pseudo-second-order model in the following formulas [36]:(2)$$\log\left(q_{e}-q_{t}\right)=\log q_{e}-\frac{k_{1}}{2.303}t$$
(3)$$\frac{t}{q_{t}}=\frac{1}{k_{2}q_{e}^{2}}+\frac{t}{q_{e}}$$
where *k*_1_ and *k*_2_ were the constants of the pseudo-first-order and pseudo-second-order adsorption rate. *q_t_* and *q_e_* (mg/g) were adsorption amount at time *t* and equilibrium time, respectively. The kinetic adsorption modal for Cu(II) were plotted and displayed in Figure 9a,b. The kinetic parameters (*k*, and *q_e_*) and correlation coefficients (*R*^2^) were calculated and showed in Table 3. The consequences showed that *R*^2^ (>0.999) values of pseudo-second-order model for both CS-L1 and CS-L3 were closer to 1, suggesting that the adsorption process was dominated by chemical adsorption, and that electrons were shared or exchanged between the adsorbate and the adsorbent [37].

Adsorption isotherms reflected the variation of adsorption amount at various adsorbate concentrations. The Freundlich and Langmuir isotherm models were applied to sketch the equilibrium process of Cu(II) sorption onto the CS membranes. The isotherm equations were expressed as follows [11]:(4)$$\frac{C_{e}}{q_{e}}=\frac{1}{q_{m}K_{L}}+\frac{C_{e}}{q_{m}}\;$$
(5)$$\log q_{e}=\frac{1}{n}\log C_{e}+\log K_{F}$$
where *q_e_* (mg/g) was the maximum adsorption amount, and *K_L_* was the Langmuir isotherm constant involved to the free energy and the binding strength. *K_F_* and *n* were Freundlich constants involved to the adsorption amount and the intensity of adsorption, respectively.

The suited lines for isotherm adsorption model were exhibited in Figure 9c,d, respectively. The relevant isotherm adsorption dates were concluded in Table 4. By comparing *R*^2^ values of CS-L1 and CS-L3, the Langmuir model (*R*^2^ > 0.99) was more suitable for equilibrium data than Freundlich model (0.9369, 0.9717), suggesting the Cu(II) adsorption showed the monolayer coverage on the adsorbent surface. In contrast, for CS-L3 stacking membranes, the *R*^2^ value of both two model were all greater than 0.95, which indicates that the multilayer adsorption behavior also acted a significant part during the adsorption process. The maximum Cu(II) adsorption amount for CS-L3 stacking membrane was 276.2 mg/g. This value was superior to most of the reported Cu(II) adsorbents in the literature (Table 5).

### 3.7. EDS Analysis

The EDS analysis and photographs of CS-L3 membrane before and after adsorption are shown in Figure 10a,b. It was observed that the mats color changed after adsorption while the morphology did not change significantly, which indicated CS-L3 membrane had excellent stability in aqueous solution. In addition, the adsorption solution was prepared by CuSO_4_ 5H_2_O solution. Compared to the pre-adsorption membrane, the new significant peaks of Cu and S appeared after adsorption (Figure 10b), which confirmed CuSO_4_ adsorbed on the CS membranes, and further indicated that the membrane had good adsorption properties.

### 3.8. Adsorption Mechanism of Cu(II) by CS Stacking Membranes

To further study the adsorption mechanisms, XPS analysis was investigated (Figure 11). In the XPS analysis of CS adsorption (Figure 11a), there were three characteristic peaks at 530.1 eV, 399.2 eV and 284.0 eV, corresponding to O1s, N1s, C1s, respectively. After adsorption, a new Cu(II) binding energy peak observed at 932.8 eV, suggesting that a large quantity of Cu(II) ions had been adsorbed onto CS membranes. For N1s spectra shown in Figure 11b,c, the N1s binding energy at 397.6 eV before adsorption shifted to 398.6 eV after adsorption, which indicated the amino group participated in the adsorption [42]. At the same time, as seen from Figure 11d,e, the characteristic peak for O1s binding energy at 531.1 eV before adsorption shifted to 531.3 eV after adsorption. The obvious shift revealed that hydroxyl groups also participated in the adsorption of Cu(II).

FT-IR spectra also demonstrated the adsorption mechanism (Figure 12). For CS stacking membrane before adsorption, the characteristic peaks at 3429 cm^−1^ and 1610 cm^−1^ belonged to hydroxyl functional group (–OH) and amino group (–NH_2_) of CS, respectively. After adsorption, a new peak appeared at 1670 cm^−1^ near the –NH_2_ and the hydroxyl peak broadened, indicating that these groups participated in the Cu(II) adsorption [24]. In addition, a new peak around 609 cm^−1^ was present, corresponding to the characteristic peaks of Cu–O, and indicating that Cu(II) existed on the CS mats surface [43]. The FT-IR results agreed with XPS analysis, suggesting that both –NH_2_ and –OH from CS membranes contributed to Cu(II) adsorption.

Based on the aforementioned study, the Cu(II) adsorption mechanism of CS stacking membranes is shown schematically in Scheme 2. The CS molecule contained a large number of amino and hydroxyl groups which could bind heavy metal ions through chelation. The main complex sites were –NH_2_ and –OH, since the N atom in –NH_2_ and the O atom in –OH had an electron pair that could be combined with Cu(II) containing empty orbitals by a coordinated covalent bond. Combined with a BET analysis of Table 2, CS stacking membranes had higher pore size, pore volumes and specific surface areas than the pure CS fiber membrane, which confirmed the CS stacking membranes possessed more adsorption sites.

## 4. Conclusions

In this research, to study the effect of hierarchical structures of a CS-based adsorbent on the adsorption behaviors of Cu(II), a CS adsorbent with a novel stacking structure was designed by alternating elelctrospinning/electrospraying. The Cu(II) adsorption performance revealed that CS stacking membranes with three layers had a maximum adsorption amount of 276.2 mg/g within 60 min at 100 mg/L for the initial concentration of Cu(II). The adsorption kinetics behavior for the CS stacking membrane was well match the pseudo-second-order model, indicating the that chemical reaction was dominated the Cu(II) adsorption process. Meanwhile, the isothermal modal for CS stacking membranes was demonstrated by both of the Freundlich and Langmuir isotherm models, which indicated the stacking structure regulated the adsorption behavior. This innovative design of the structure of the natural adsorbent may provide guidance for further research on the competitive adsorption of various heavy metal ions and the application of natural, environmentally-friendly polymer materials as adsorbents.

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
