# Peer review of "Preparation of Chitosan Stacking Membranes for Adsorption of Copper Ions"

_polymers, 2019, doi:10.3390/polym11091463_

Round 1
Reviewer 1 Report
The 'Preparation of chitosan stacking membranes for adsorption of copper ions' manuscript yields good results and can be accepted for publication in Polymers after some adjustments:
1.Check in abstract: ... "CS fibers sandwiched CS micro-hemispheres" ...
2.Replace in all text M with mol/L.
3.Insert in topic 2.5 data regarding the surface area measurement technique.
4. Check how Equation 1 is presented.
5. In Table 1, why does such a significant difference occur between HFIP / FA = 9/1 and 2wt% CS?
6. Adsorption tests with chitosan fibers, hemispheres and power are required to confirm membrane development using two different preparation techniques.
7. In Figure 6a it is necessary to show the standard deviation to confirm the sequence shown on line 215, since there is apparently no difference between the curves.
8. Check which pH of copper precipitation in the form of hydroxide, as from pH 6 this may already be occurring (Figure 7).
9.Why CS-L5 material was not used in figure 8?
10. It is necessary to propose an interaction scheme between copper and adsorbent material!
Reviewer 2 Report
The manuscript by Zhang and co-workers concerns the preparation and characterization of a new polymer membrane, based on chitosan, able to adsorb copper ions, in view of its possible application wastewater remediation. The membrane present a complex architecture alternating layers of electrospun fibers and electrosprayed microparticles, all of chitosan. A variety of techniques is used for the membrane characterization, including IR, SEM, XPS. Copper adsorption is also investigated in some details.
In principle, the subject is relevant and worth to be investigated. Polymer membranes represent a suitable tool to reduce the contaminant concentration in polluted water, with evident ecological interest. The manuscript is well structured: the introduction is exhaustive and clearly introduces the aims, experimental section is quite complete, data presentation is clear and conclusions are reliable (even though a deeper discussion should be done, see below). I support publication of this manuscript in Polymers once the following points will be considered by the authors, and the manuscript amended accordingly.
Major points:
In some parts, the manuscript is written more as a technical report than as a scientific contribution. Efforts can be done in better rationalizing the outcomes of the experiments, deepening their interpretation. As an example, subsection 3.1 seems more a part of a technical report than a part of a scientific article. Some discussion is needed, introducing the choice of the solvents. Moreover, the mechanism underpinning the Cu ions adsorption on the membrane should be interpreted on a molecular basis, also explaining the reasons for which the data better fit a second order kinetics. This would enrich the paper, also enlarging its interest for a wider readership.
All the quantitative values and the uncertainties are not correctly quoted. As an example 48.6 +/- 12.5 should be 49 +/- 12, and 126.9 +/- 23.8 should be 127 +/- 2.
The diameter of the fibers obtained from HFIP/FA with volume ratio of 9/1 appears quite polydisperse, from Fig. 1f; so, I wonder if the average value is reliable. This point requires further discussion.
Indeed, the increase of Cu adsorption due to the stacked architecture of the membrane is quite limited. My feeling is that the authors claims should be more moderate on this point. It seems the electrospun chitosan is per se a good adsorbent. This point should be better discussed by the authors.
In Fig. 10b the S signal has to be better commented.
interpretation of FT-IR spectra in Fig. 12 is really poor.
In the conclusions, the authors should better highlight the advances this work brings in the field, possibly envisaging directions for future research.
Does the membrane present any selectivity for Cu ions? Why other polluting ions were not considered?
The text contains a certain number of errors and clumsy phrases. In this respect, the manuscript must be carefully amended before it could be published. Below, I list just some of the needed corrections:
p.1 line 18: "which dominated by CS" should be "which are dominated by the CS"
p.1 line 25: "an universal heavy metal contamination in wastewater " should be "a heavy metal universally contaminating wastewater" or something similar.
p.1 line 34: "various od adsorbents" should be "various adsorbents.
p.2 line 60: "for" should be "of"
p.2 line 63: "the thickness" should be "the effects of the thickness"
p.2 line 65: delete "of"
p.2 line 85: "for 5 h by electrospinning " shuld be "by electrospinning for 5 h"
p.3 line 108: delete "ranged"
p.3 eq. 1: check the format. The same holds for all the equations.
p.6 line 174: "The Stability in Aqueous Solution".
p.6 line 180: "be" should be "derive from"
